# Modelling parameter uncertainty reveals bushmeat yields versus survival trade-offs in heavily-hunted duiker *Cephalophus* spp.

**Tatsiana Barychka**[1]*, **Drew W. Purves**[2]◉, **E. J. Milner-Gulland**[3], **Georgina M. Mace**[1]◉

**1** Centre for Biodiversity and Environment Research, Department of Genetics, Evolution and Environment, University College London, London, United Kingdom, **2** DeepMind, London, United Kingdom, **3** Department of Zoology, University of Oxford, Oxford, Oxfordshire, United Kingdom

◉ These authors contributed equally to this work.
* tatsiana.barychka.14@ucl.ac.uk

**Data Availability Statement:** All relevant data are within the manuscript and its Supporting Information files.

## Abstract

Reliably predicting sustainable exploitation levels for many tropical species subject to hunting remains a difficult task, largely because of the inherent uncertainty associated with estimating parameters related to both population dynamics and hunting pressure. Here, we investigate a modelling approach to support decisions in bushmeat management which explicitly considers parameter uncertainty. We apply the approach to duiker *Cephalophus* spp., assuming either a constant quota-based, or a constant proportional harvesting, strategy. Within each strategy, we evaluate different hunting levels in terms of both average yield and survival probability, over different time horizons. Under quota-based harvesting, considering uncertainty revealed a trade-off between yield and extinction probability that was not evident when ignoring uncertainty. The highest yield was returned by a quota that implied a 40% extinction risk, whereas limiting extinction risk to 10% reduced yield by 50%-70%. By contrast, under proportional harvesting, there was no trade-off between yield and extinction probability. The maximum proportion returned a yield comparable with the maximum possible under quota-based harvesting, but with extinction risk below 10%. However, proportional harvesting can be harder to implement in practice because it depends on an estimate of population size. In both harvesting approaches, predicted yields were highly right-skewed with median yields differing from mean yields, implying that decision outcomes depend on attitude to risk. The analysis shows how an explicit consideration of all available information, including uncertainty, can, as part of a wider process involving multiple stakeholders, help inform harvesting policies.

## Introduction

Many studies raise alarm over the present rate of wild meat harvesting as a major cause of population decline and extinction risk for many species [1–3]. With wild meat providing a major source of protein and household income to some of the world's poorest people [4–6], both subsistence and commercial hunting in West and Central Africa are on the rise [1, 6, 7].

**Funding:** T.B. was funded by the Natural Environment Research Council (NERC), grant number NE/L002485/1, https://nerc.ukri.org/. The NERC had no role in study design, data collection and analysis, decision to publish, or preparation of the manuscript. DeepMind (DeepMind Technologies Limited, 6 Pancras Square, London, N1C 4AG, UK) provided support in the form of salaries for D.P., but did not have any additional role in the study design, data collection and analysis, decision to publish, or preparation of the manuscript. The specific roles of D.P. are articulated in the 'author contributions' section.

**Competing interests:** D.P. received salary from DeepMind (DeepMind Technologies Limited, 6 Pancras Square, London, N1C 4AG, UK). This commercial affiliation does not alter our adherence to PLOS ONE policies on sharing data and materials. The authors have no potentially competing interests to declare. The submission is not related to any patents, patent applications, or products in development or for market.

Bushmeat harvest across the Congo Basin alone is estimated to occur at more than six times the sustainable harvest rate [8].

However, reliably estimating a sustainable harvest level remains problematic. Ecological systems are highly complex and the relevant biological data on mammals in tropical forests is scarce [9]. Information is often collected during short field seasons [10–12], across different spatial scales and in different ecosystems [13, 14], producing point estimates of population parameters and species abundances that vary considerably between studies [15, 16]. As a result, traditional techniques such as monitoring offtakes and correlating them with changes in harvested species dynamics such as abundance and age structure [15, 17] will not accurately assess the sustainability of harvesting. To address this problem, a number of approaches have been developed ranging from the relatively simple Robinson and Redford index [18] to the more sophisticated Bayesian techniques used in fisheries [19, 20]. Instead of using time-series data on animal densities and offtakes, these methods take point estimates of populations' carrying capacity and rate of population growth as inputs. This allows an estimation of sustainable levels of production of harvested populations [15] which can then be compared with actual data on animal offtakes. However, to be effective these methods still require accurate estimates of population parameters, and even then the simple indices can be misleading [21]. As a result, the suggested sustainable harvest levels could differ substantially from the actual sustainable levels, but the extent of this mismatch is unknown. In response, the general recommendation is to adjust harvest rates downwards to reduce chances of a human-caused mortality going above a limit that could lead to the depletion of the population [22]. But without an explicit consideration of uncertainty there is no objective way to set the size of this adjustment [23], potentially leading to harvest levels that could still be high risk for the bushmeat species involved, or unnecessarily limiting the bushmeat yield available to local populations.

In this study, we introduce a method for calculating sustainable harvesting levels based on an explicit treatment of parameter uncertainty in harvesting models. Outcomes are evaluated in terms of survival probability and yield, and the level of uncertainty of that yield. We examine the results for two constant harvesting strategies (quota-based and proportional) and over a number of harvesting time horizons.

We illustrate our method with a case study of duiker harvesting in sub-Saharan Africa. Duikers are widely harvested in Central Africa, contributing over 75% of the harvested bushmeat in Central African Republic and Cameroon [14, 24]. Compared to other bushmeat species (e.g. primates, pigs, rodents) duikers are relatively well-studied: there are multiple published estimates of population parameters [3, 25–27]. However, as population estimates vary considerably between studies (Van Vliet and Nasi [16] demonstrated a four times difference in estimates of population growth rates for *Cephalophus monticola* from two methods), true parameter values are unknown. This implies that ignoring uncertainty could be highly misleading and calls for an approach that considers the uncertainty explicitly. We apply our uncertainty-based method to duiker *Cephalophus* spp., but it can potentially be used to estimate sustainable harvest rates for any data-deficient exploited species.

## Materials and methods

### Modelling population dynamics

**Population model.** We begin by describing our modelling approach. To model population dynamics, we used the Beverton-Holt population model [28].

$$N_{t+1} = \frac{r_t N_t}{1 + [(r_t - 1)/K]N_t} \tag{1}$$

where $N_t$ is the population density (individuals per unit area: in this case, animals km$^{-2}$) at time $t$; $N_{t+1}$ is the population density in the following time step; $K$ is the equilibrium population size in the absence of harvesting; and $r_t$ is the density-independent intrinsic rate of natural increase (the balance of births and deaths) for year $t$.

The Beverton-Holt model has been widely used in the past to study the dynamics of harvested species [4, 29]. We chose to use it because it is compensatory rather than over-compensatory [30] and it is believed to provide a robust representation of intraspecies competition in ungulate populations that are not constrained by resources or habitat availability [31]. Both these properties make it suitable for characterising duiker dynamics.

The year-to-year fluctuation in births and deaths (i.e. environmental stochasticity) was represented by varying $r$ between years, as follows:

$$r_t \sim \mathbb{N}\{r, \sigma\} \tag{2}$$

where $r_t$ was the value of $r$ that applied in simulation year $t$, $r = exp(r_{max})$ and $\sigma$ was the standard deviation for $r$ across all years. Following methods by Lande, Sæther and Engen [32], we assumed a coefficient of variation of 0.10, implying $\sigma = 0.10 \times r$, with 0.10 being the lowest value implemented by Lande, Sæther and Engen [32] reflecting low climate variability in the tropics.

**Model parameterisation: Prior belief.** Parameters $r_{max}$ and $K$ were supplied to the Beverton-Holt population model with uncertainty in these parameters, as follows:

For each of the two parameters $r_{max}$ and $K$ in the population model, we drew from a prior distribution reflecting beliefs about the likely distribution of values of the parameter (based on our empirical duiker dataset, see Field Data below and S1 Table in S1 File), i.e. we assumed that a true value of $r_{max}$ applied to a given local population, but we assumed also that this value was unknown. Hence, we use a probability distribution for $r_{max}$, which reflects our degree of belief in the likely values based on field data.

As the prior for $r_{max}$, we used a log-normal distribution as follows:

$$ln\,(r_{max}) \sim \mathbb{N}\{\widehat{r}_{max}, \tilde{r}_{max}\} \tag{3}$$

where $\widehat{r}_{max}$ is the mean of log-transformed values of $r_{max}$ (4) established from field data, and $\tilde{r}_{max}$ is the standard deviation of the log-transformed values of $r_{max}$ (5). A log-normal distribution was used instead of a normal distribution (also see S1 Fig and S1 Appendix in S1 File) to constrain $r_{max}$ and $K$ to positive values. We assumed that the log-transformed values of the reported parameter values in our field data were independent samples from the distributions defined in (3). The simplest approach was then to set $\widehat{r}_{max}$ and $\tilde{r}_{max}$ using the field data as follows:

$$\widehat{r}_{max} = mean\{ln\,(r_{max}^{data})\} \tag{4}$$

$$\tilde{r}_{max} = sd\{ln\,(r_{max}^{data})\} \tag{5}$$

where $r_{max}^{data}$ denotes the values of $r_{max}$ reported in the data.

A small value of $\tilde{r}_{max}$ implied that, based on field data, we were highly certain that the true value of $r_{max}$ was very close to $\widehat{r}_{max}$. A large value of $\tilde{r}_{max}$ implied that we were highly uncertain about the true value of $r_{max}$, such that it could lay a long way from $\widehat{r}_{max}$. More precisely, the choice of $\tilde{r}_{max}$ implied that we were 95% certain that the true value of $r_{max}$ was in the range $exp\,\{ln\,(\widehat{r}_{max}) - 1.96\tilde{r}_{max}\}$ and $exp\,\{ln\,(\widehat{r}_{max}) + 1.96\tilde{r}_{max}\}$.

Likewise, we used as the prior for $K$:

$$ln\,(K) \sim \mathbb{N}\{\widehat{K}, \tilde{K}\} \tag{6}$$

where $\widehat{K}$ was the mean of $K$ (defined using field data) and $\tilde{K}$ was the standard deviation of the log-transformed values of $K$.

The sampled prior distributions for $r_{max}$ and $K$, along with the empirical estimates (field data), are presented in S1 Fig in S1 File.

In addition to analysis with parameter uncertainty, we ran simulations without parameter uncertainty (but with environmental stochasticity), to provide a baseline comparison. For simulations without parameter uncertainty, we used the mean values of $r_{max}$ and $K$ only (i.e. $\widehat{r}_{max}$ and $\widehat{K}$) based on field data for each duiker species, to parameterise the Beverton-Holt population model.

**Harvesting strategies.** To implement a simple, reasonable, harvesting strategy, we assumed that harvesting occurred at a constant rate, set as either a quota or proportional to population size. That is, each year, a quota $h$ or a proportion $\varphi$ of the population was targeted, and this target did not vary among years ((7) and (8), respectively).

$$N_{t+1} = \frac{r_t N_t}{1 + [(r_t - 1)/K]N_t} - h \tag{7}$$

$$N_{t+1} = \frac{r_t N_t}{1 + [(r_t - 1)/K]N_t} - \varphi N_t \tag{8}$$

Note that the proportion $\varphi$ is an aggregate parameter of harvesting effort and could in practice be altered by changing the number of hunting days per year, the density of traps, the efficacy of traps used, the proportion of animals released after being trapped, the proportion of land set aside as reserve, and so on; $h$ is simply the number of animals removed per year.

Total population losses to harvesting, or yield ($Y_t$) at time $t$, is the difference between the number of animals at time $t$ after reproduction at the end of year $t-1$ (equation (13) in S1 Appendix in S1 File), and the higher of 0 and the number of surviving animals after the target quota/proportion has been applied (equations (12) and (13) in S1 Appendix in S1 File).

**Simulation experiment.** For each duiker species we simulated quota-based and proportional harvesting over a 25-year harvest period. Based on model estimates, we assessed average yields, survival probability, and the uncertainty in both yield and survival, over five 5-year increments.

For proportional harvesting, we examined values of $\varphi$ from 0 (no harvest) to 0.90 in discrete steps of 0.05, giving 19 different values of $\varphi$. For quota-based harvesting, the ranges of target quotas $h$ for each species were found experimentally, by running harvesting simulations with an increasing upper limit to $h$ ($0 \leq h \leq 13$) and examining summary statistics (mean yield, median yield and mean survival probability) from harvesting each species over 50 years. This resulted in target quota ranges of between: 0 and 3.5 animals km$^{-2}$ year$^{-1}$ for Peters' duiker, 0 and 1.5 animals km$^{-2}$ year$^{-1}$ for bay duiker, and 0 and 10 animals km$^{-2}$ year$^{-1}$ for blue duiker. We included zero-rate harvesting in both proportional and quota-based harvesting simulations to create a baseline scenario. The initial population size $N_0$ was set randomly, by drawing from a uniform distribution between $0.20K$ and $0.80K$.

For each harvest rate we carried out an ensemble of 1000 simulations. Harvesting was applied from year 1 onwards (no harvesting took place in year 0). The ensemble size was based on preliminary analysis involving comparing summary statistics and visualising results for smaller (100 simulations and 500 simulations) and larger (10000 simulations) sample sizes. For each simulation within each ensemble, we drew a value for each parameter at random from the prior.

Survival probability was equal to the proportion of simulations without quasi-extinction. Quasi-extinction was defined as the population density dropping below 0.1 animals km$^{-2}$ at any point during the simulation, based on the lower end of density estimates collected in areas of high harvesting intensity [33, 34]. A response of 1 was assigned to a year where population size $N_t$ was equal to or was above a threshold of 0.1 animals km$^{-2}$; zero (0) was assigned to a year (and all the following years) where population size dipped below the viability threshold (after which we set $N_t$ to zero in all future years). Responses were then averaged to give an estimate of survival probability at each harvest rate with 95% confidence intervals over 5-year harvests. A detailed description of our method is presented in S1 Appendix in S1 File.

In addition to the 25-year harvesting, we simulated quota-based and proportional harvesting over a range of harvesting horizons (100, 50, 20 and 5 years) for each duiker species. For each combination of timescale (100, 50, 20 and 5 years) and harvest rate, we carried out an ensemble of 1000 simulations and estimated bushmeat yields and species survival probability. All simulations were run in R version 3.6.3 [35]. Results are reported with one standard deviation.

## Field data

**Parameter estimates.** Three *Cephalophus* species: Peters' duiker *C. callipygus*, bay duiker *C. dorsalis* and blue duiker *C. monticola* (also known under the scientific name *Philantomba monticola*) [36, 37] were selected as our case study based on availability of independent and published empirical estimates of population parameters and their relative importance for wild meat supply in sub-Saharan Africa [38]. Candidate studies were identified using Google Scholar and Web of Science (using search terms: bushmeat, wild meat, tropical, Africa), and by searching the cited references in the collated papers. The following selection criteria were used to prioritise studies from which data were gathered: (a) pertaining to one of the three duiker species; (b) meeting basic quality requirements, i.e. we discarded studies where the method for estimating parameters was not specified; and (c) containing primary data which could be used to inform the calculation of either of the two key parameters: intrinsic rate of population increase (the maximal growth rate) $r_{max}$ and carrying capacity $K$, where $K$ was the number of animals per km$^2$ estimated in un-hunted sites. The parameter estimates were combined into a duiker dataset (S1 Table in S1 File).

Where available, estimates of population growth rate were taken directly from the literature. Alternatively, we used one of two methods—Cole's [39] and Caughley and Krebs [40] (S2 Appendix in S1 File)—to estimate $r_{max}$ based on information provided by the authors (such as body mass ranges for the three duiker species). In addition, as an independent test of whether the estimates of $K$ were reasonable, the allometric estimates of population density at $K$ for the three duikers were also calculated, based on the relationship between population density and body mass for mammalian primary consumers described by Damuth [41]:

$$D = a(log\ W) + b \tag{9}$$

where $D$ is the population density, $W$ is the duiker body mass in grams, $a$ = - 0.75 is the slope of the relationship and $b$ = 4.23 is the estimated intercept.

**Actual bushmeat offtakes.** We estimated bushmeat offtakes for the three duiker antelope species using estimates of total bushmeat exploitation for the Congo basin [38, 42] (S3 Appendix in S1 File).

**Framework summary.** Two measures of harvesting outcome were used in our decision framework: expected yield and probability of species survival. The choice of harvesting strategy was motivated by maximising expected bushmeat yield over the duration of harvesting

horizon. With reference to species survival probability, we used a minimum survival threshold of 90% of population [43] over the duration of harvesting horizon as a benchmark. The optimum harvesting strategy was the strategy that maximised yield subject to a survival probability constraint. The summary workflow is presented in Fig 1 and S4 Appendix in S1 File.

## Results

### Duiker dataset

We identified and assessed twenty six potential sources of primary data on population parameters $r_{max}$ and $K$, including two PhD theses [10, 38]. Parameter estimates from the thirteen studies that met our selection criteria were combined into a dataset of carrying capacity, $K$ and intrinsic rate of natural increase, $r_{max}$, for our three duiker species. S1 Table in S1 File gives the observed values for $r_{max}$ and $K$. The mean values for model parameters $r_{max}$ and $K$ ($\mu_{rmax}$ and $\mu_K$), and the variability of estimates (standard deviations, $s_{rmax}$ and $s_K$), along with average body masses and sample sizes for each of the three duiker species in our dataset, are given in Table 1. The spatial distribution of field studies is presented in Fig 2.

Geographically, the studies were concentrated in five main research areas: the Ituri Forest (Democratic Republic of Congo); Makokou (north-eastern Gabon); Bioko and Rio Muno (Cameroon); Dzanga-Sangha and Dzanga-Ndoki National Parks (Central African Republic), and Arabuko Sokoke (Kenya). The areas from which our dataset came were between 160 kilometers and 3500 kilometers apart; at least 100 times the size of known duiker ranges [10]. The east-west spread of samples in our dataset may explain some of the variation in parameter values (due to habitat and environmental differences). However, parameter estimates still varied substantially within the areas where more than one estimate was available; e.g. estimates for the density of blue duikers in un-hunted areas (our proxy for carrying capacity) ranged between 10.2 [34] and 61 [11] animals km$^{-2}$ in the Ituri Forest, DRC. Overall, Peters' duiker was the most difficult to find data on. Most estimates of carrying capacity dated from the late 1970s-80s, with the latest estimates in 2000 [3, 34].

### Baseline predictions ignoring uncertainty

As a baseline against which to compare our main results, we examined predictions from a model in which we ignored uncertainty. The choice of optimum harvesting level was comparatively easy for proportional harvesting, because the maximum harvest rate resulted in 100% survival for all three species. For quota-based harvesting, the harvesting strategy that maximised predicted yield (which we refer to as the maximum harvesting rate) also resulted in a 100% predicted survival probability for all species except for bay duiker (Table 2). For all three species and under both harvesting strategies (quota-based and proportional), the models predicted that average yield peaked at intermediate harvesting levels and the probability of population survival declined with increasing harvesting levels, but only after the maximum yield had already been exceeded (Figs 3–5, S5-S7 Figs in S1 File).

### Predictions considering uncertainty

Predictions considering uncertainty revealed a hitherto hidden trade-off between yield and survival. For quota-based harvesting, the predicted maximum harvesting rates were similar to those generated from the baseline (Table 2). However, the predicted survival at this harvesting rate was much lower. Constraining the harvesting to achieve a predicted survival of at least 90% resulted in much lower yields (Figs 6–8, S5-S7 Figs in S1 File).

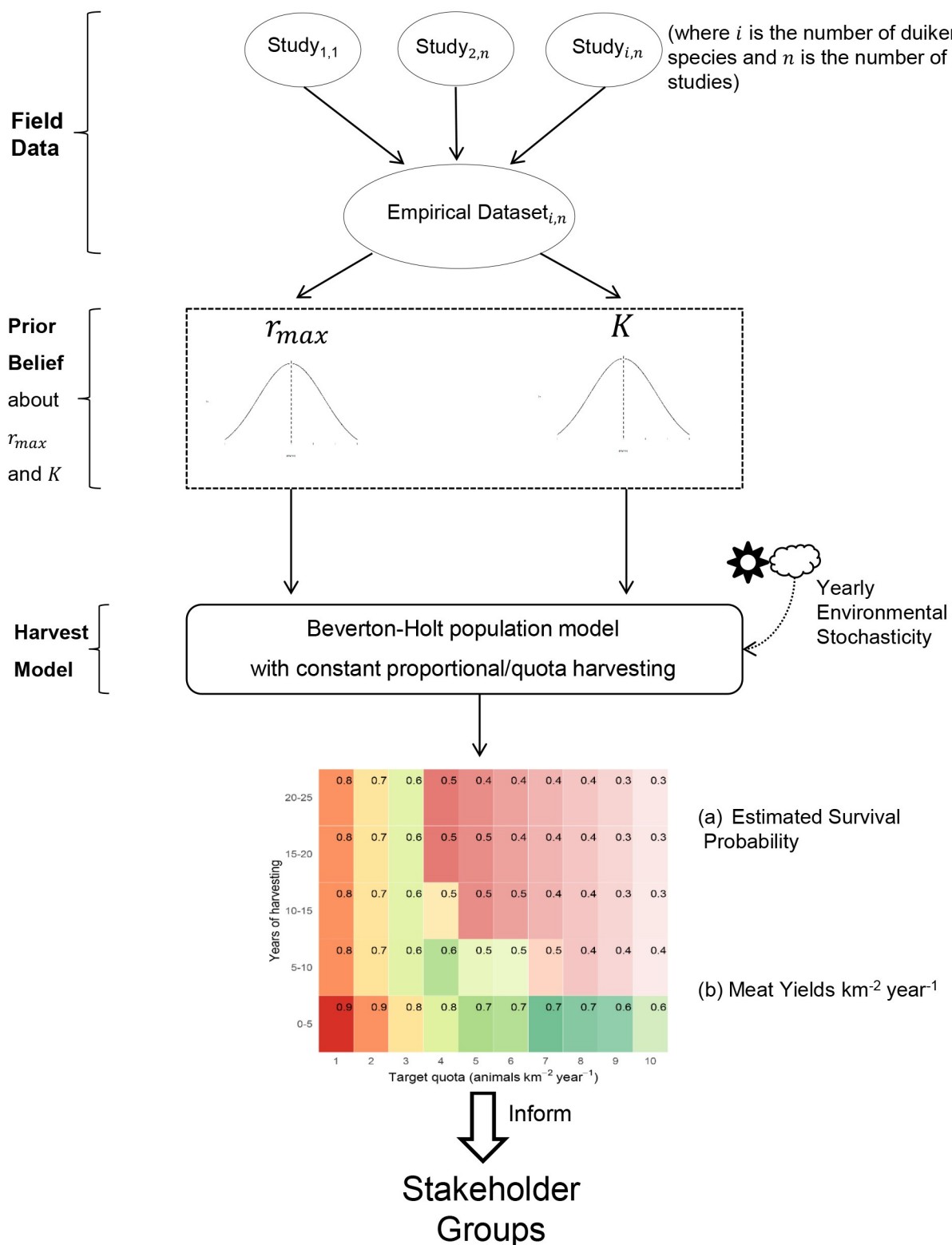

**Fig 1. Our method of combining field data with the harvest model.**

**Table 1. Estimates of population parameters $r_{max}$ and $K$ for Peters' duiker *C. callipygus*, bay duiker *C. dorsalis* and blue duiker *C. monticola*.**

| Species | Body Mass $\mu(s)$ | $r_{max}$ | | | $K$ | | | |
|---|---|---|---|---|---|---|---|---|
| | | $n$ | $\mu_{rmax}$ | $s_{rmax}$ | $n$ | $\mu_K$ | $s_K$ | Allometric[1] |
| *C. callipygus* | 16.22 (2.60) | 5 | 0.44 | 0.14 | 4 | 9.70 | 3.62 | 11.82 |
| *C. dorsalis* | 17.99 (2.83) | 6 | 0.39 | 0.14 | 6 | 5.43 | 2.55 | 10.96 |
| *C. monticola* | 4.62 (0.55) | 7 | 0.58 | 0.27 | 7 | 39.46 | 26.72 | 30.31 |

Mean ($\mu$) $r_{max}$ and $K$, sample size ($n$) and body mass estimates, with 1 standard deviation ($s$) based on field data.

[1]Density $D = a(\log W) + b$ (9), where $W$ is the duiker body mass in grams, $a = -0.75$ is the slope of the relationship and $b = 4.23$ is the estimated intercept [41].

Predictions for proportional harvesting (S5-S7 Figs in S1 File) shared key features with the predictions for quota-based harvesting. However, declines in survival probability and average yields after the maximum harvest rate (the rate that maximised yield) were noticeably more gradual under proportional harvesting than under quota-based harvesting. Maximum proportional yields were comparable with maximum quota-based yields (Table 2). However, unlike quota-based harvesting, survival remained above 90% at maximum harvest rates, compared to 50%-60% survival under the maximum quota.

Assessing the impacts of harvesting over longer harvesting horizons, i.e. beyond the first 5 years of harvesting, was clearly important for optimal harvesting, as aiming to maximise yields in the first years led to species extinctions. In all cases, including parameter uncertainty in the

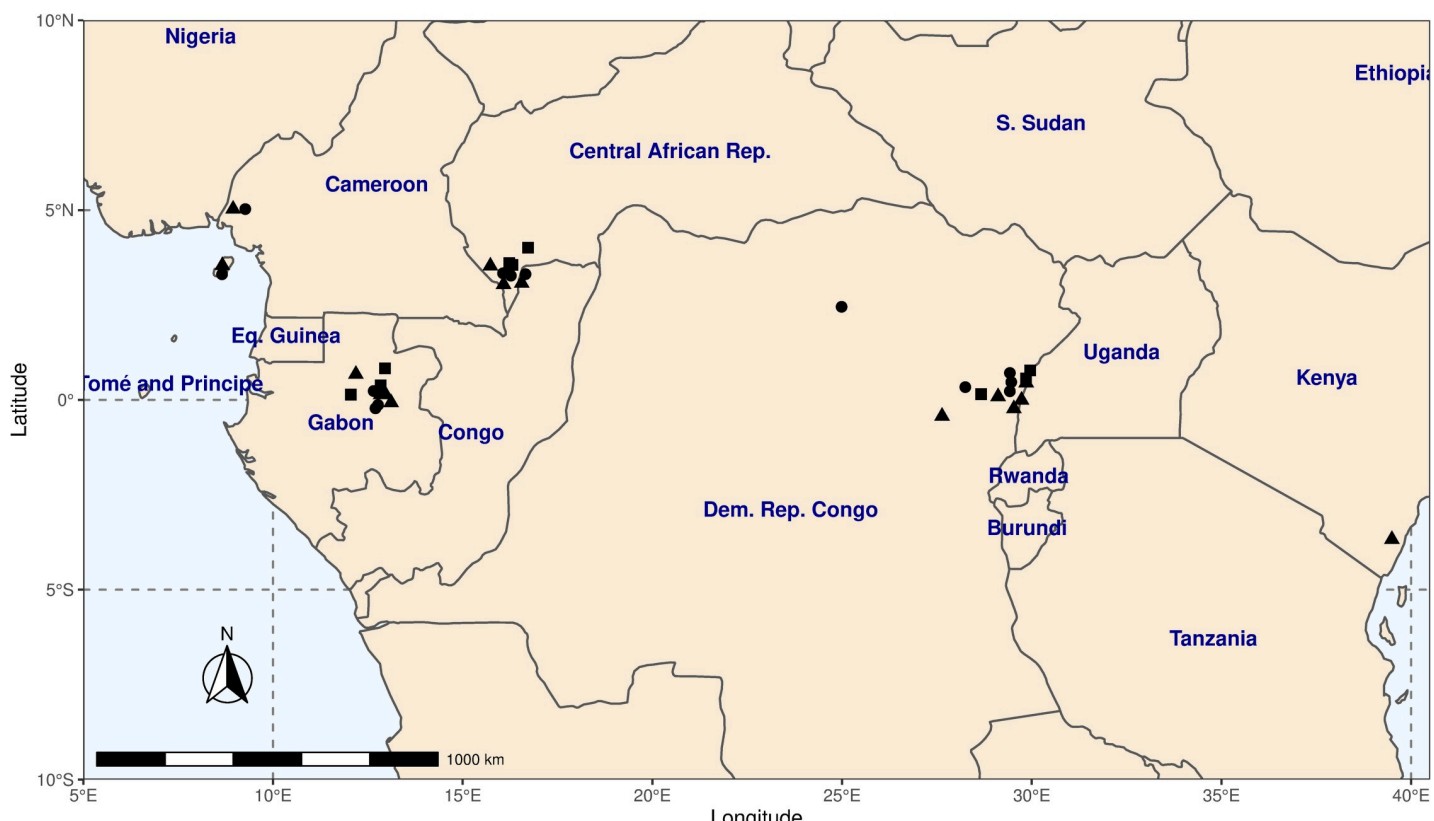

**Fig 2. Geographic locations of field studies included in the duiker dataset.** Studies comprised of Peters' duiker *C. callipygus* (squares), bay duiker *C. dorsalis* (circles) and blue duiker *C. monticola* (triangles). Made with Natural Earth. Free vector and raster map data @ naturalearthdata.com.

**Table 2. Maximum harvest rates from harvesting duiker *C. callipygus*, *C. dorsalis* and *C. monticola* with (wU) and without (nU) parameter uncertainty.**

| Species | Harvest Rate | | Yield (s) | | | | Survival Probability (s) | | | |
|---|---|---|---|---|---|---|---|---|---|---|
| | Q | P | Q | | P | | Q | | P | |
| | | | nU | wU | nU | wU | nU | wU | nU | wU |
| *C. callipygus* | 1 | 0.2 | 1.0 (0.2) | 1.0 (0.49) | 0.96 (0.21) | 0.75 (6.31) | 1.0 (0) | 0.61 (0.49) | 1.0 (0) | 0.98 (0.14) |
| *C. dorsalis* | 0.5 | 0.2 | 0.5 (0.25) | 0.5 (0.25) | 0.42 (0.1) | 0.33 (5.57) | 0.6 (0.5) | 0.55 (0.5) | 1.0 (0) | 0.93 (0.26) |
| *C. monticola* | 4 | 0.2 | 4.0 (0.8) | 4.0 (1.98) | 3.84 (0.79) | 2.67 (116.8) | 1.0 (0) | 0.56 (0.5) | 1.0 (0) | 0.99 (0.11) |

Quota animals per km² per year (Q), proportion of the population (P), estimated yields (animals km⁻² year⁻¹) and associated survival probabilities, with 1 standard deviation (s).

harvesting model did not change the maximum feasible rate of harvesting. However, it exposed the risk to species survival, particularly at intermediate harvest rates.

With parameter uncertainty for a given species, harvesting approach, and harvesting level, there tended to be a large amount of uncertainty in the predictions, most notably for mean yield, where standard deviations were in some cases greater than the median (Table 2). Moreover, the estimated yields for a given harvesting level were often highly right-skewed, with

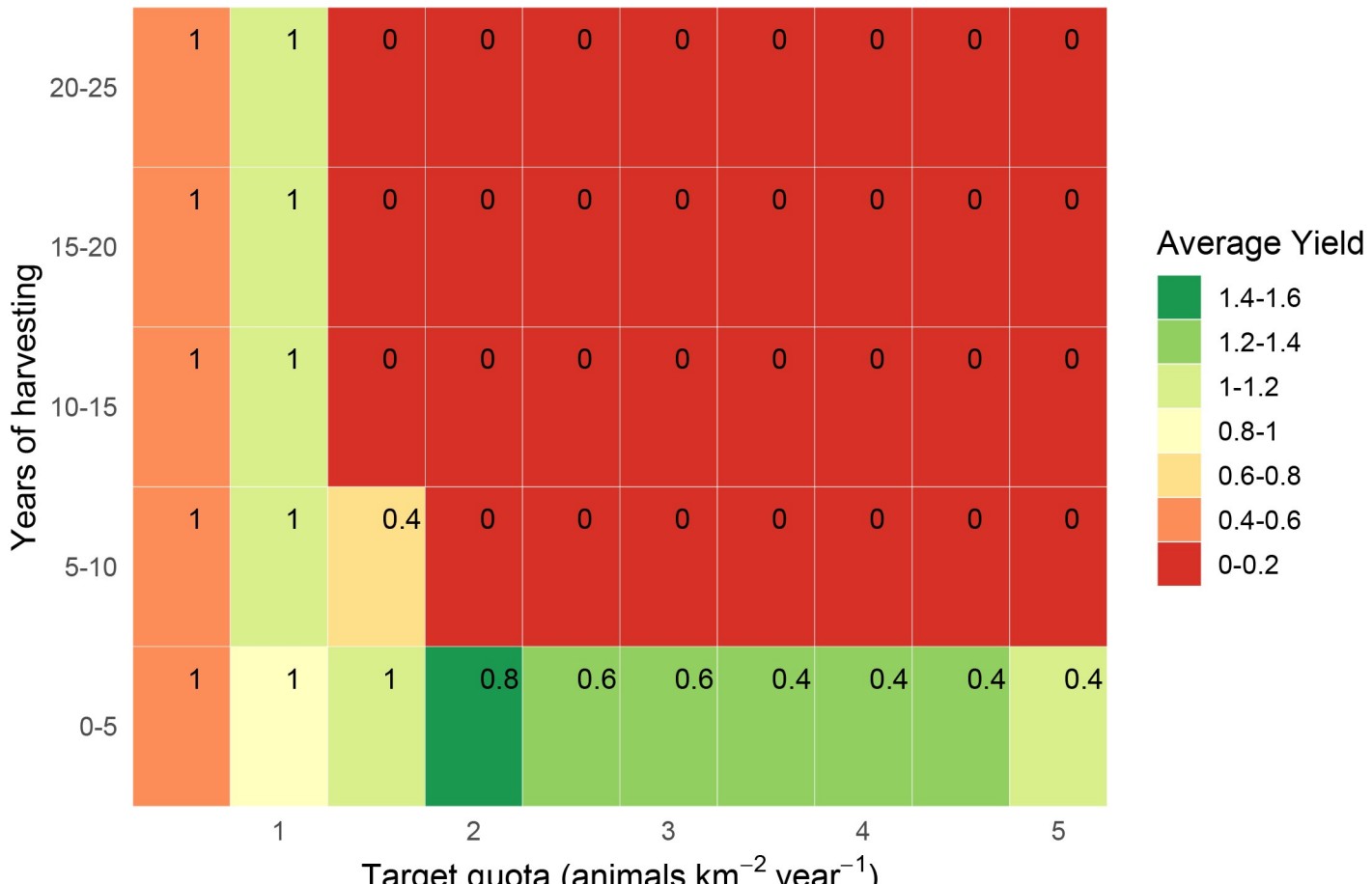

**Fig 3. Estimated yields (animals km⁻² year⁻¹) from quota-based harvesting of Peters' duiker *C. callipygus*.** Yields are estimated over 25 years in 5-year increments, without parameter uncertainty, with corresponding survival probabilities (in top-right corner of each rectangle).

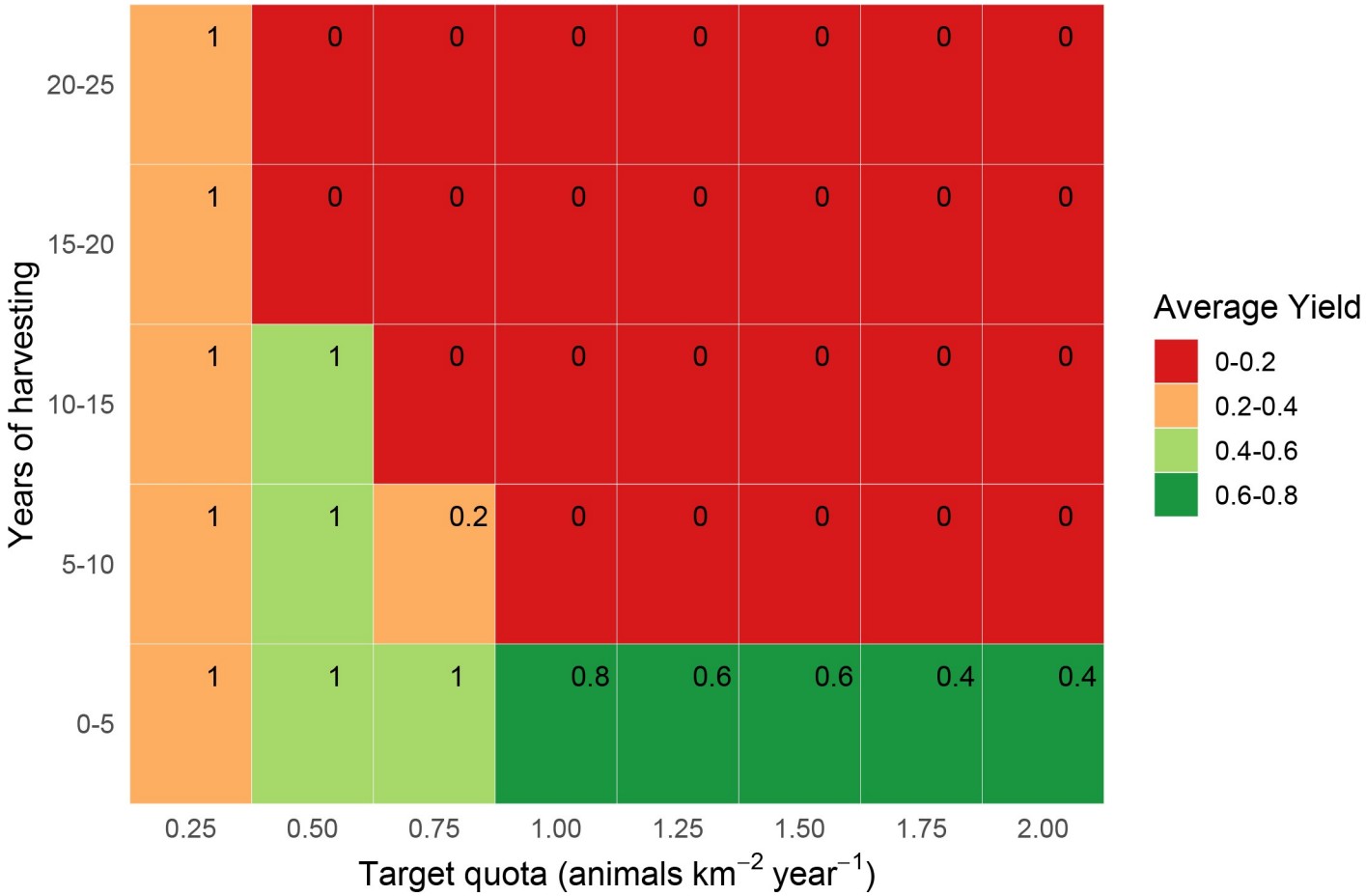

**Fig 4. Estimated yields (animals km$^{-2}$ year$^{-1}$) from quota-based harvesting of bay duiker *C. dorsalis*.** Yields are estimated over 25 years in 5-year increments, without parameter uncertainty, with corresponding survival probabilities (in top-right corner of each rectangle).

most predictions for each ensemble returning yields somewhat below the median, and a small number of simulations returning yields much greater than the median (S2-S4 Figs in S1 File). As a result of the uncertainty and the skew, the harvesting level that maximized the median yield, was often very different to the levels maximizing the mean yield, or yields in the 1$^{st}$ or 3$^{rd}$ quartiles (S2-S4 Figs in S1 File). The right skew was particularly high at medium-to-high harvest rates; we therefore used the median rather than the mean as a yield statistic for all harvesting scenarios under uncertainty (Figs 6–8, Table 2 and S5-S7 Figs in S1 File).

Against these generalities, there were important differences by species, harvesting method, and time horizon, as discussed below.

### Peters' duiker: Quota-based harvesting

For Peters' duiker, we estimated a maximum quota-based yield of 1–1.2 animals km$^{-2}$ year$^{-1}$ (Figs 3 and 6), which corresponded well with recorded bushmeat offtakes for Peters' duiker (S3 Appendix in S1 File). The maximum harvesting level resulted in a 100% population survival without uncertainty (Fig 3) and in 50%-80% survival when uncertainty was included (Fig 6). With parameter uncertainty, achieving 90% target survival involved a reduction in yield of over 50% (harvest level ≤ 0.5 animals km$^{-2}$ year$^{-1}$). Harvesting ≥1.5 animals km$^{-2}$ year$^{-1}$

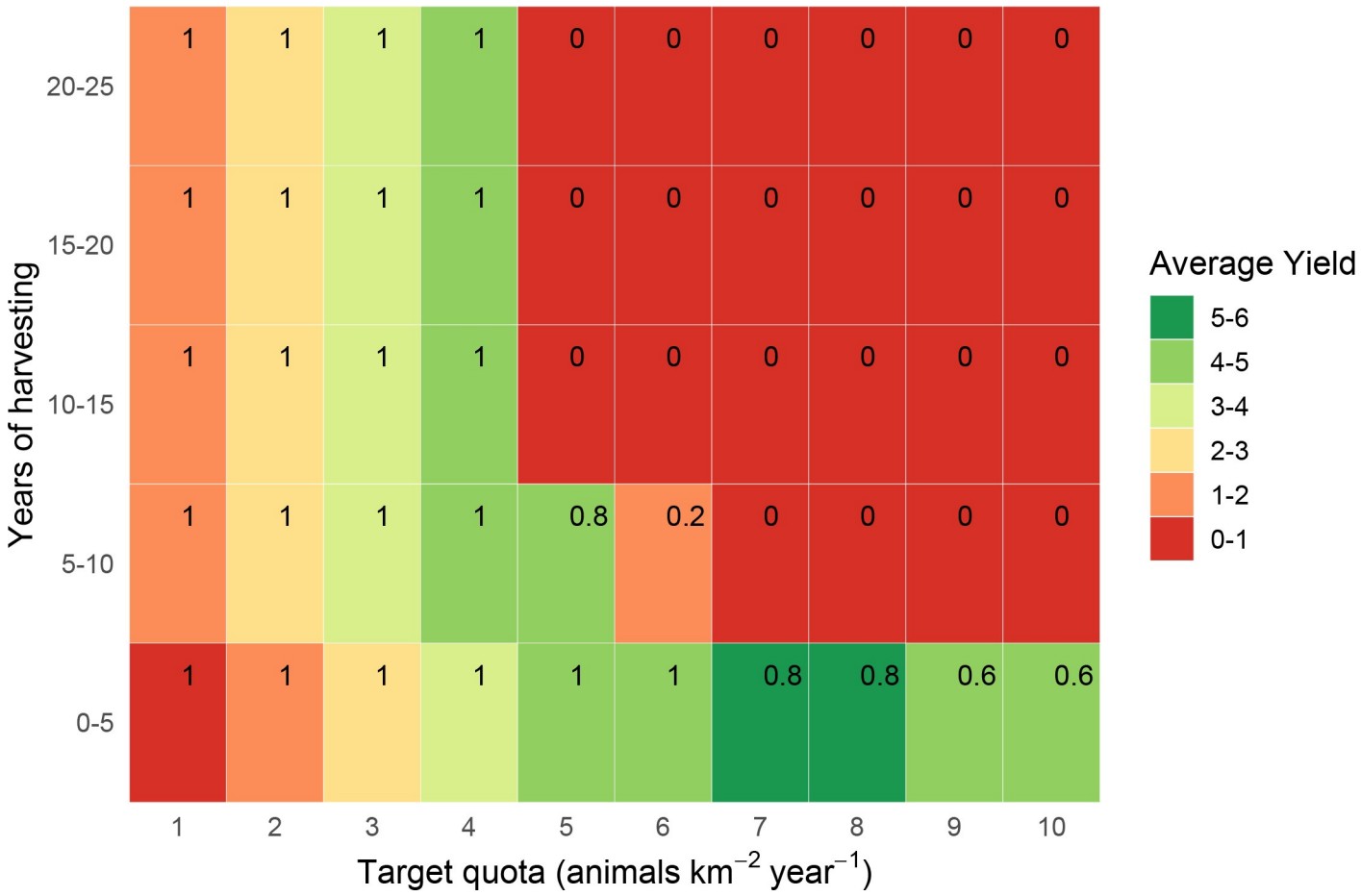

**Fig 5. Estimated yields (animals km$^{-2}$ year$^{-1}$) from quota-based harvesting of blue duiker *C. monticola*.** Yields are estimated over 25 years in 5-year increments, without parameter uncertainty, with corresponding survival probabilities (in top-right corner of each rectangle).

resulted in 50%-90% extinction probability for all combinations of target quota and time interval.

### Peters' duiker: Proportional harvesting

As for quota-based harvesting, proportional harvesting yielded 1–1.2 Peters' duiker km$^{-2}$ year$^{-1}$ at the maximum harvest rate (S5 Fig in S1 File). Unlike quota-based harvesting however, survival probability at the maximum harvest rate was high (90%-100%) even when parameter uncertainty was included (S5 Fig in S1 File). With parameter uncertainty, yields were lower on average than without and were highly variable, with an average standard deviation of 6.31 animals km$^{-2}$ year$^{-1}$ under maximum harvesting (Table 2). Unlike the quota-based strategy, proportional harvesting at intermediate harvest rates (20%-30% of duiker population km$^{-2}$ year$^{-1}$) maintained sustainable animal populations (survival ≥80%) even beyond the first 5 years.

### Bay and blue duiker: Quota-based harvesting

The estimates for quota-based harvesting for bay and blue duiker were qualitatively similar to those for Peters' duiker, but there were important quantitative differences (Table 2, Figs 4, 5, 7 and 8). For the same time horizon, bay duiker had a lower maximum yield than Peters', and

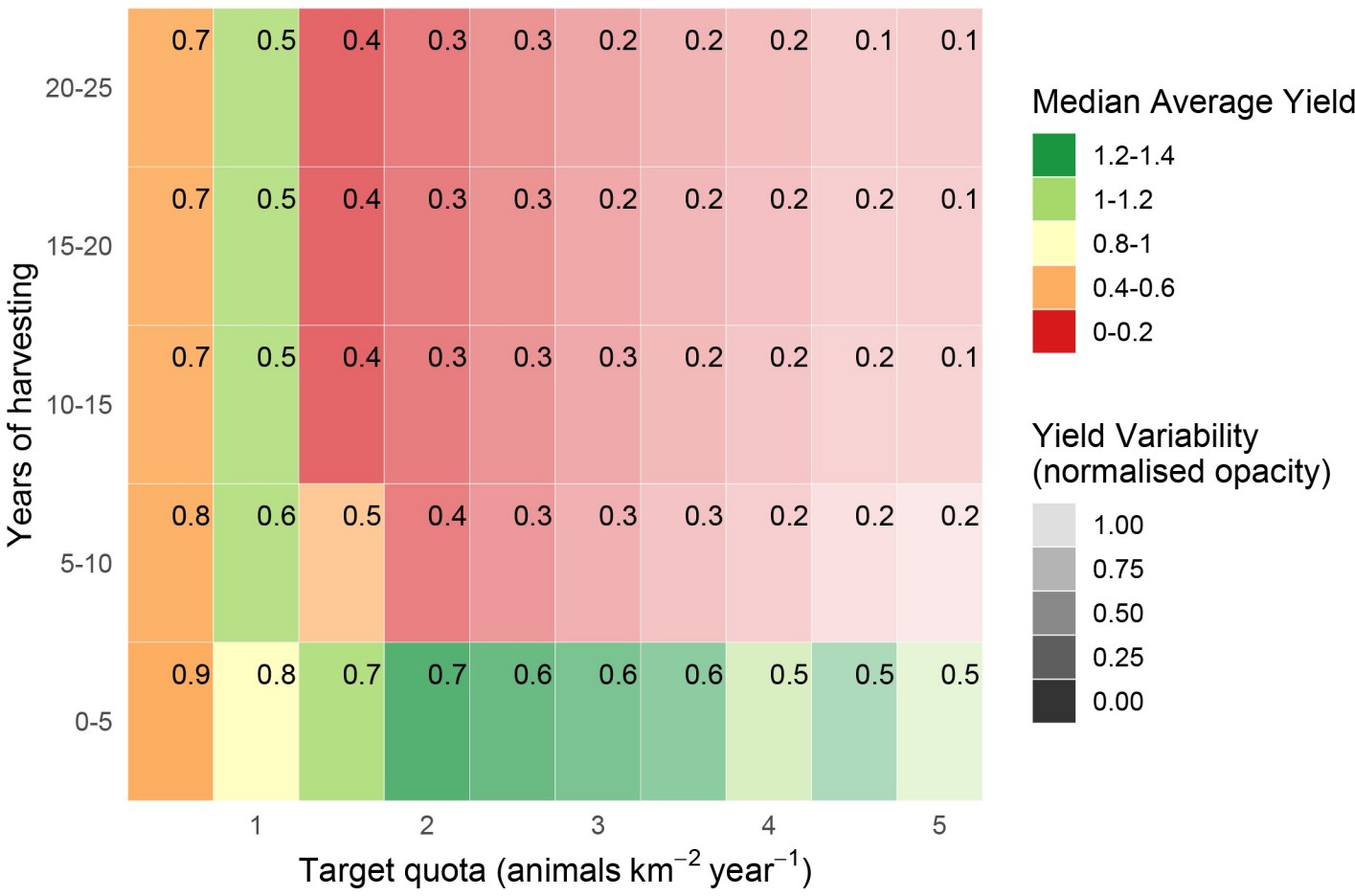

**Fig 6. Estimated yields (animals km⁻² year⁻¹) from quota-based harvesting of Peters' duiker *C. callipygus*.** Yields are estimated in 5-year increments over 25 years, with parameter uncertainty, with corresponding survival probabilities (in top-right corner of each rectangle).

blue had a higher yield than both Peters' and bay duiker: 1–1.2 animals km⁻² year⁻¹ (Peters'), 0.4–0.6 animals km⁻² year⁻¹ (bay), and 4–5 animals km⁻² year⁻¹ (blue). At and just above the maximum harvesting level (target quotas of 0.5–0.75 animals km² year⁻¹ for bay, and 4–5 animals km² year⁻¹ for blue duiker), the risk of extinction was estimated to be about 50%-60% for both bay and blue duiker beyond the first 5 years of harvesting (Figs 7 and 8). Maximum yields from the model were significantly lower than the recorded offtakes of 2.62–5.02 bay duiker km⁻² year⁻¹ and 14.47–25.39 blue duiker km⁻² year⁻¹ (S3 Appendix in S1 File).

Harvesting conservatively yielded 0.2–0.4 bay duiker (Fig 7) and 1–3 blue duiker km⁻² year⁻¹ (Fig 8). For blue duiker, the more conservative harvesting (1 blue duiker km⁻² year⁻¹) resulted in a 67% reduction in yield compared to the maximum, with an increase in survival probability to 80%-90% under quota-based policy.

The uncertainty of predictions was greatest for blue duiker (Table 2). With harvest rates well above sustainable levels (for example, at $h \geq 5$ in Fig 8), yields from blue duiker may remain high in the short term despite overharvesting. The prediction for population survival vs harvesting level was also closer to linear under quota-based harvesting (S4 Fig in S1 File). This further complicates decision making, because with a relationship closer to linear, the exact choice of harvest rate has a larger impact on the quota and yield. For example, a less-

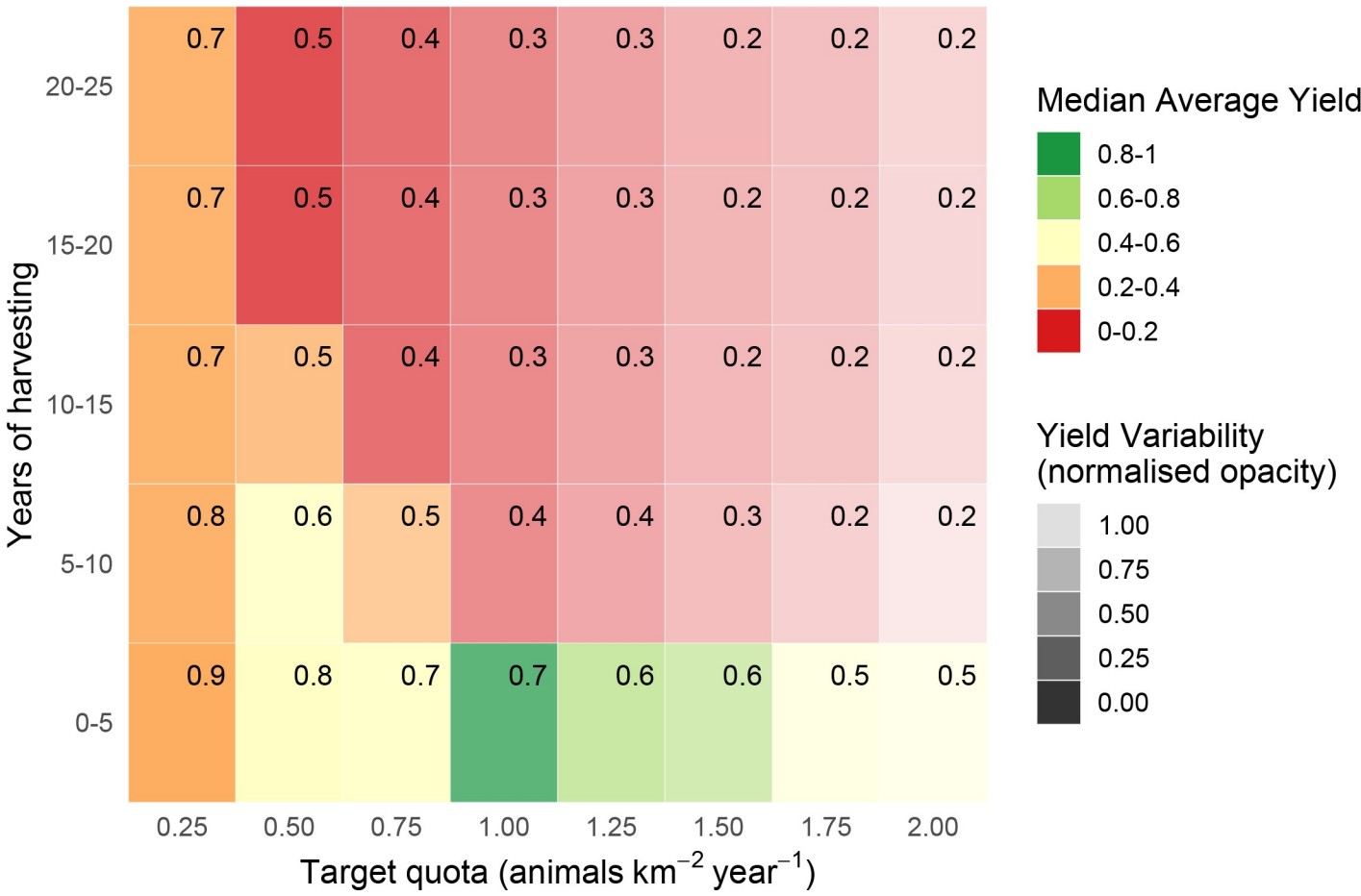

**Fig 7. Estimated yields (animals km$^{-2}$ year$^{-1}$) from quota-based harvesting of bay duiker *C. dorsalis*.** Yields are estimated over 25 years in 5-year increments, with parameter uncertainty, with corresponding survival probabilities (in top-right corner of each rectangle).

conservative decision-maker targeting around 10 blue duiker km$^{-2}$ year$^{-1}$ could cause an extinction risk of 60% over 100 years (S4 Fig in S1 File).

## Bay and blue duiker: Proportional harvesting

The maximum yields under proportional harvesting were noticeably lower for bay than for Peters' duiker: 0.4–0.6 animal km$^{-2}$ year$^{-1}$ without parameter uncertainty, decreasing marginally to 0.2–0.4 animal km$^{-2}$ year$^{-1}$ when parameter uncertainty was introduced (S6 Fig in S1 File). The threshold at which harvesting decreased survival was also lower, with a statistically significant effect being seen when 30% of the population was harvested (S6 Fig in S1 File).

Harvesting conservatively at 10% of the population size yielded 0.2–0.4 bay duikers km$^{-2}$ year$^{-1}$ (S6 Fig in S1 File)–a yield reduction of 30%-50% compared to the maximum. At these low rates, extinctions were comparatively rare (100% survival, on average) and yield variability was relatively low, suggesting that population was growing despite harvesting. Proportional strategies were more sensitive to overharvesting for bay than for Peters' duiker; however, still less so than quota-based harvesting.

The maximum yields were significantly higher and with very high variability (*s* = 116.80 with uncertainty, Table 2) for blue duiker than for Peters' and bay duiker, reflecting higher densities and population growth rates. Under a proportional harvesting strategy, the estimated

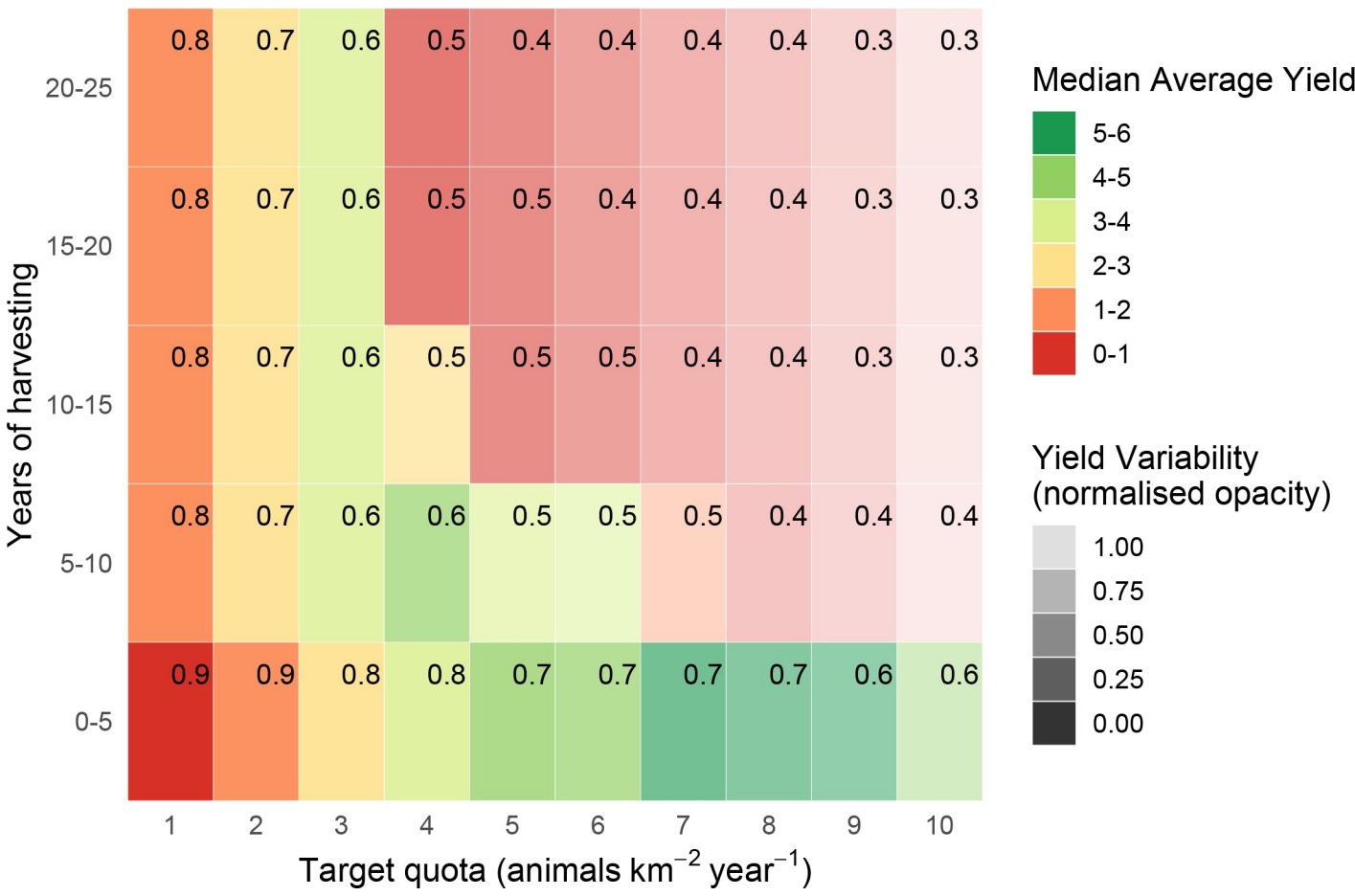

**Fig 8. Estimated yields (animals km$^{-2}$ year$^{-1}$) from quota-based harvesting of blue duiker *C. monticola*.** Yields are estimated over 25 years in 5-year increments, with parameter uncertainty, with corresponding survival probabilities (in top-right corner of each rectangle).

yields were maximised at a harvest rate of 20%-30% of the blue duiker population (S7 Fig in S1 File) with little difference between years in terms of survival. Extracting 30% of the population resulted in an average survival probability of between 0.8 and 1. Without considering uncertainty, harvesting up to 40% of the population returned a survival probability of 1.

## Discussion

Our analysis demonstrates significant potential benefits of incorporating parameter uncertainty into model-based analyses of sustainable bushmeat yields. All such model-based analyses [4, 15, 16] can only ever form part of the complex decision process that eventually leads to harvesting practice on the ground [6, 44–46]. However, the incorporation of uncertainty does reveal some key features that may inform the stakeholders that influence harvesting. In particular, for quota-based harvesting, considering parameter uncertainty reveals an important trade-off between yield and population survival; reveals highly uncertain and skewed outcomes for any given policy; and makes the idea of proportional harvesting seem all the more attractive compared to quota-based harvesting.

The trade-off between yield and survival is absent, or much reduced, in analyses ignoring uncertainty, where the choice of optimum harvesting may appear simple because harvesting that maximises yield also maximises survival probability. To understand why, consider that

the uncertainty-based analysis deals with an ensemble of model simulations, which can be thought of as a set of populations with different parameters. An analysis ignoring uncertainty effectively deals with just one of these populations, in which case the policy that maximises yield needs to keep the population extant for most of the harvesting period. In contrast, in the presence of uncertainty, a policy can maximise overall yield by setting a quota that harvests effectively from the most productive populations, at the cost of increased extinction risk for the less productive populations. This is also true for proportional harvesting; however, under proportional harvesting, only a share of animals is ever extracted, and this share is proportionally lower in less productive populations than in more productive populations (see below). This means that even when harvest rates are set too high (for example, due to imperfect knowledge of a local population), under proportional harvesting part of the population survives whereas every animal might be extracted under quota-based harvesting. However, if overharvesting continues, under proportional as well as quota-based harvesting, some populations can eventually become unviable (represented here by the 0.1 animals $km^{-2}$ extinction threshold) leading to local extinctions.

Our approach begins by acknowledging that our knowledge of species is not perfect [15, 16], as demonstrated here for our duiker antelope species. In addition, animal populations are subject to demographic and environmental variability [29, 47–49]. Lack of species data, as well as natural variability, are major sources of uncertainty about real-life populations and their responses to harvesting [16]. Our analysis shows that considering parameter uncertainty for quota-based harvesting [50] could have major impacts on decision-making. Most notably, considering uncertainty revealed a stark trade-off between yield and survival for all three species. Those policies that maximised yield resulted in low survival rates (0.61, 0.55 and 0.56 for Peters', bay and blue duiker, respectively; Table 2), whereas policies constrained to ensure high survival rates resulted in much lower yields (50%-70% yield reduction).

For a given harvesting policy, we also found highly variable, right-skewed predicted yields. For the maximum yield, the standard deviation on yield was often over 100% of the mean (e.g. for blue duiker, yield = 2.67 animals $km^{-2}$ $year^{-1}$ and standard deviation = 116.80). This is despite the fact that we chose this set of species specifically because they were relatively well studied [10, 13, 51, 52]. As a result of the uncertainty in yield, the apparent best policy was highly dependent on decision makers' attitude to risk [53], especially for quota-based harvesting. The importance of the uncertainty in yield also depends in part on scale. If the parameter variation occurs at fine scales, then stakeholders can expect yields that average over the distributions. However, if the parameters vary coarsely, then the analysis implies that a given stakeholder may receive a yield that is very different from the average. The skewed nature of the distributions implies further that for every stakeholder lucky enough to gain substantially more than the average, there would be many receiving substantially less–a situation of few winners and many losers. This observation could be potentially important in weighing up the economic implications of harvesting at local or regional scales.

Finally, our analysis showed that proportional harvesting was much more robust to uncertainty than quota-based harvesting. This is not a new result [49, 54, 55]. Based on likely ranges for the duikers' reproduction rates and population densities (S1 Fig in S1 File), proportional harvesting showed a reduced trade-off between yield and survival, and a greater survival probability for a given average yield (S5-S7 Figs in S1 File). Proportional harvesting brings two main potential benefits in terms of the survival of local populations. First, proportional harvesting naturally adjusts the number of animals taken year to year, such that in years with unusual low population densities, fewer animals are taken. Second, as mentioned above, proportional harvesting naturally removes fewer animals from those local populations with lower carrying capacities, lower growth rates, or both. In our analysis, the first benefit was apparent

in both our baseline case, and our main analysis; but the second benefit was only revealed in our main analysis, when parameter uncertainty was considered. Thus, proportional harvesting can return higher yields overall, whilst keeping more of the vulnerable populations extant [32]. The analysis shows that proportional harvesting is not perfect in this regard (the policy that maximises yield still results in some extinctions). Therefore, we caution against using the maximum target proportion. But, in this theoretical analysis, it clearly outperforms quota-based harvesting—even more so than it does in the baseline case.

However, it is important to recognise that despite its obvious benefits proportional harvesting is often considered to be unfeasible in Central Africa [50]. In principle, harvesting could be limited to a proportion of animal densities by, for example, keeping the number of snares constant. However, this is not always feasible due to poor harvesting regulation in West and Central Africa. A potential compromise might be to use proportional thinking to explicitly set dynamic local quotas [56–58]. Whether, when and how the potential, theoretical benefits of proportional harvesting can be translated into benefits for real bushmeat harvesting remains to be seen.

According to our model, blue duiker was the most high-yielding species (yields as high as 4 animals $km^{-2}$ $year^{-1}$, $s = 0.8–1.98$), followed by Peters' duiker (up to 1 animal $km^{-2}$ $year^{-1}$, $s = 0.2–0.49$) and bay duiker (0.5 animals $km^{-2}$ $year^{-1}$, $s = 0.25$). Out of the three species, bay duiker was particularly sensitive to harvesting, with optimal target offtakes as low as 0.25 animals $km^{-2}$ $year^{-1}$ (Figs 4 and 7), i.e. 1 duiker per 20 $km^2$ $year^{-1}$. The maximum target quotas were noticeably higher over a shorter time horizon (5 to 10 years). For example, for Peters' duiker, the short-term (0–5 years; Figs 3 and 6) vs longer-term (10–25 years) target quota rates increased nearly three-fold: from 0.5–1 animal $km^{-2}$ $year^{-1}$ to 2 animals $km^{-2}$ $year^{-1}$. However, if a 5-year harvesting horizon was used to set harvest targets, long-term species survival probability dropped to around 39% (Fig 6).

Under proportional harvesting, the maximum harvest rate of 20% annually was surprisingly consistent across species, but was higher on average than the sustainable harvest rates suggested by Noss [14] of 1.2%-12.8%, 1.6%-12.8% and 2.3%-17.2% for Peters', bay and blue duiker, respectively. Our modelled estimates at the maximum yield were comparable with the least conservative sustainable offtakes calculated by Noss [27] using Robinson and Redford's formula, and with Payne's [10] estimates in Korup National Park, Cameroon [59]. When harvesting conservatively (i.e. limiting harvest rates to ensure 90% survival over a 100-year harvesting horizon, S2 Fig in S1 File), our optimal yields were lower, and closer to Noss's [27] most conservative estimates. This degree of agreement between our analysis and the independent analysis of Noss [12, 14] is encouraging. However, actual reported offtakes (S3 Appendix in S1 File) are greater than our predicted sustainable yields for two of the three duiker species, and similar for Peters duiker (Table 3), which is worrying in terms of current sustainability.

Large ranges around the predicted yields in our model may be explained by the fact that, unlike most studies [39, 52], we used a range of estimates of $K$ to parameterise the harvesting system. These estimates of carrying capacity were quite variable, for example, ranging from 10.2 blue duikers $km^{-2}$ in the Ituri Forest, north-eastern Democratic Republic of Congo [25] to around 70 blue duikers $km^{-2}$ in north-eastern Gabon estimated by Feer [60]. The reasons for these discrepancies could be manifold: different measuring techniques [13], observation error [61–63], or a spatial gradient as suggested by Peres [64] in his comparison of hunted and non-hunted sites across the Amazonian rain forest. This makes cross-habitat generalisations about optimal harvesting rates more difficult. Unfortunately, our sample sizes were not sufficient to explore the mechanisms underlying variations in empirically-based estimates of $K$ in more detail.

**Table 3. Modelled bushmeat yields (animals km-2 year-1) for Peters' duiker *C. callipygus*, bay duiker *C. dorsalis* and blue duiker *C. monticola*.**

| Species | Sustainable yields (animals km$^{-2}$ year$^{-1}$) | | Optimal yields (animals km$^{-2}$ year$^{-1}$) | Actual offtakes (animals km$^{-2}$ year$^{-1}$) |
|---|---|---|---|---|
| | Noss [27] | Payne [10] | Our model | see S3 Appendix in S1 File |
| *C. callipygus* | 0.01–2.09 | - | 0.1–1.3 | 1.00 |
| *C. dorsalis* | 0.003–1.17 | 0.16–0.33 | 0.05–0.6 | 2.62–5.02 |
| *C. monticola* | 0.24–10.50 | 2.38–4.18 | 0.1–2.4 | 14.47–25.39 |

Compared to sustainable bushmeat yield estimates by Noss [3] and Payne [10] and actual bushmeat offtakes (S3 Appendix in S1 File). For optimal yields, survival probability was ≥0.90 over 100 years (S2-S4 Figs in S1 File).

Like all models, ours is a simplification of real-life processes. Firstly, harvesting rates vary between years [65]. However, by examining survival and yields over different timeframes and harvesting strategies this work presents a novel and a useful perspective on wild meat harvesting under uncertainty. Secondly, using a relatively simple analytical model such as the Beverton-Holt model provides certain advantages over stochastic simulation studies [53, 66, 67], such as more generalisable, robust conclusions that capture the most salient features of population dynamics useful for exploring system sensitivity to different parameter values and guiding more detailed simulation studies of particular situations [32, 65]. Other population models could easily be used instead of the Beverton-Holt model [68], and employing different models would allow model uncertainty (ignored here) to be addressed. More sophisticated harvesting policies such as threshold harvesting policies [69], or no-take reserves are sometimes feasible [70, 71]; however in most cases, and certainly in West and Central Africa, managers have relatively little control over resource users and harvest intensities. Thirdly, the values for population growth rates and carrying capacity were sampled from a log-normal distribution which has a relatively large right tail, leading to a larger variance of the estimates. More reliable estimates of population growth rates and carrying capacity are therefore critical to enable more precise predictions. Finally, we did not account for the likely replenishment of the vacant areas (i.e. areas were duikers had been exhausted) by immigrants from the surrounding unhunted populations (i.e. source-sink structure). Assuming that immigration/emigration can occur, the extinction risk should be lower than we predicted; however, given the pervading uncertainty, we recommend erring on the side of caution.

Here, we developed a relatively simple model-based approach for informing decisions in bushmeat harvesting under high parameter uncertainty. High parameter uncertainty is common in the tropics. Although we used the duikers to illustrate our approach, the approach can be used to help inform management decisions for any harvested species. The need to translate theoretical research into practical solutions which can facilitate decision-making in conservation has been widely recognised [72–74] and a diverse range of tools is now available, in particular in marine conservation [75, 76] and in spatial planning and prioritisation [77, 78]. Recognising the need to make our modelling approach more accessible to bushmeat practitioners, we also built an online interactive application [40]. A screen shot of our online application is presented in S8 Fig in S1 File. Practical implementations of conservation actions based on applications of modelling techniques are still relatively rare [76]. With further improvements, more sophisticated interactive decision-support tools can be developed, ideally with input from bushmeat practitioners.

Given that bushmeat is an essential source of protein and additional income for many of the poorest people in West and Central Africa, the potential for improvements in bushmeat yields, species survival probability and predictability of yields should be explored using

adaptive management within a participatory setting where local people are active participants in management planning about their own resources [79].

## Conclusion

Here, we explored the potential impact of considering uncertainty when seeking sustainable bushmeat harvesting policies. Considering uncertainty revealed trade-offs resulting from quota-based and proportional harvesting of three duiker *Cephalophus* spp. under realistic conditions of parameter uncertainty. The uncertainty was quantified using empirical data, explicitly modelled and used to inform a decision framework that we developed. Although our model could not eliminate uncertainty, by handling it in a systematic and transparent way [80, 81], it helped identify the potential impacts of uncertain parameters on decision-making [53, 82], laying out boundaries for sustainable harvesting. It is obviously preferable to use data to set prior beliefs wherever possible [83, 84]. However, even in the absence of any data, it may still be possible to define reasonable priors on parameters based on expert judgement [80]. Such priors could still be used with our method, and we would argue that doing so would be better than not using modelling at all, or using modelling but ignoring uncertainty.

The socioeconomic reality of bushmeat harvesting is such that harvesting levels would rarely be set by any single quantitative algorithm. Combining different techniques, such as the population modelling used here with trend analysis, could result in more reliable assessments of sustainability of bushmeat harvesting for data-deficient species. Importantly for bushmeat, the process should involve stakeholders at all scales: local people, resource extraction companies, local and state government authorities and scientists [6]. We used duiker antelope *Cephalophus* spp. as a case study. However, in principle, the uncertainty- and risk-based method introduced here could be applied to any harvested species and could, as part of a wider process involving multiple stakeholders, help place bushmeat hunting on a more sustainable footing.

## Supporting information

**S1 File.**
(DOCX)

## Acknowledgments

We would like to thank Dr. Tim Newbold, Prof. Luca Borger and Dr. Matthew Holden for their comments on the manuscript. We are also grateful to Dr. Daniel Ingram and Prof. Jorn Scharlemann for their insights and suggestions.

## Author Contributions

**Conceptualization:** Tatsiana Barychka, Drew W. Purves, E. J. Milner-Gulland, Georgina M. Mace.

**Data curation:** Tatsiana Barychka.

**Formal analysis:** Tatsiana Barychka.

**Investigation:** Tatsiana Barychka.

**Methodology:** Tatsiana Barychka, Drew W. Purves, E. J. Milner-Gulland, Georgina M. Mace.

**Project administration:** Tatsiana Barychka, Drew W. Purves.

**Resources:** Tatsiana Barychka.

**Supervision:** Drew W. Purves, Georgina M. Mace.

**Validation:** Tatsiana Barychka.

**Visualization:** Tatsiana Barychka.

**Writing – original draft:** Tatsiana Barychka.

**Writing – review & editing:** Tatsiana Barychka, Drew W. Purves, E. J. Milner-Gulland, Georgina M. Mace.

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
