## [Decision Letter · Decision Letter 0]

20 Jul 2020

PONE-D-20-16287

Modelling parameter uncertainty reveals bushmeat yield versus survival trade-offs in heavily-hunted duiker Cephalophus spp.

PLOS ONE

Dear Dr. Barychka,

Thank you for submitting your manuscript to PLOS ONE. After careful consideration, we feel that it has merit but does not fully meet PLOS ONE’s publication criteria as it currently stands. Therefore, we invite you to submit a revised version of the manuscript that addresses the points raised during the review process.

Your manuscript presents an important contribution to the literature on bushmeat harvest, though the methods are also applicable in other harvest contexts as well as other modeling estimation contexts. Below you will find comments from two reviewers that should help in your revisions. 

We look forward to receiving your revised manuscript.

Kind regards,

Stephanie S. Romanach, Ph.D.

Academic Editor

PLOS ONE

Journal Requirements:

'

T.B. was funded by the Natural Environment Research Council (NERC), grant number NE/L002485/1, https://nerc.ukri.org/. The funders had no role in study design, data collection and analysis, decision to publish, or preparation of the manuscript.'

We note that one or more of the authors are employed by a commercial company: DeepMind

3. We note that Figure 1 and Figure S13 in your submission contain map images which may be copyrighted.

a. You may seek permission from the original copyright holder of Figure 1 and Figure S13 to publish the content specifically under the CC BY 4.0 license. 

Reviewers' comments:

Reviewer's Responses to Questions

**Comments to the Author**

1. Is the manuscript technically sound, and do the data support the conclusions?

Reviewer #1: Yes

Reviewer #2: Yes

2. Has the statistical analysis been performed appropriately and rigorously? 

Reviewer #1: Yes

Reviewer #2: Yes

3. Have the authors made all data underlying the findings in their manuscript fully available?

Reviewer #1: Yes

Reviewer #2: Yes

4. Is the manuscript presented in an intelligible fashion and written in standard English?

Reviewer #1: Yes

Reviewer #2: Yes

5. Review Comments to the Author

Reviewer #1: Dear authors

Congratulations for a nicely designed robust study on the impact of uncertainties in the estimation of sustainable offtake for duiker species. I believe this is a very good contribution to the body of research on sustainable hunting as well as something that could be used by practitioners for other species or places.

I have no major comments to make on the paper, the methods are well explain and statistically sound to me and the conclusions reflect faithfully the findings. Useful comparisons are provided with existing studies and show the value of the proposed approach.

This is beyond the scope of the present paper but I think the next step should be to look at the "scale" issue and how hunting is done. This might have a strong influence on expected sustainable yield. You hint on these issues L178-L181 about the many ways harvesting effort can be altered and L434-441 about the issue of scale. I am wondering to which extant it would be interesting to try selecting a very well know species (e.g. roe deer) and test these....

A few notes about references,

[1] is probably not the best one to quote for showing harvesting greater than six times the sustainable harvest rate. Maybe it would be better to refer to the latest analyses provided in https://doi.org/10.17528/cifor/007046

[27,31] are a bit outdated in terms of offtakes too and more recent references could be found in https://doi.org/10.17528/cifor/007046

This of course doesn't change the validity of the paper and of the conclusions

Reviewer #2: This paper addresses an important issue for the sustainable management of data-deficient animals. In the actual world, parameters of animal population dynamics can stochastically vary, and the uncertainty often leads to a different scenario of population dynamics. The authors explicitly incorporate the uncertainty of the parameters (r and K) in the Beverton-Holt population model and apply the model to three species of duikers in Central Africa. The authors revealed that, under quota-based harvesting, there may be a trade-off between yield and extinction probability, which was not evident when not accounting for the uncertainty of the parameters.

The manuscript is well-written and contains sufficient interest and originality to merit publication in PLOS One. However, there are several constraints and issues that should be mentioned explicitly as below.

Major comments

1. The approach presented in the manuscript can be applied to any species, as mentioned in the text. I think this point should be stressed more explicitly. For example, the model details should be explained before mentioning the procedures of actual application to the duikers (in Method section). It is better to regard the application to the duikers as a case study using the more-widely applicable approaches.

2. The approach in the manuscript still has limitations to predict bushmeat yield and extinction probability, although I admit that the authors tried to do the best within the constraints. The authors should specifically mention the limitations in the Discussion. In particular, the following two points should be mentioned.

The model assumes that the population is spatially closed. However, in real world in central Africa, local people often harvest bushmeat in specific area near village within a large continuous forest. In such environments, the vacant area (i.e. the area where duikers are exhausted) should be rapidly occupied by the surrounding unhunted populations (i.e. source-sink structure). Assuming that immigration/emigration can occur, the extinction risk should be much lower than the results of the simulations.

The results of the simulations in the manuscript should largely dependent on the distribution that the parameter, r and K follow. The author assumes that the mean and sd of these parameters follow a log-normal distribution. This distribution is relatively fat-tailed, inevitably leading to a larger variance of the estimates. To make more precise predictions, it is critical to accumulate more reliable information on r and K in more fields.

3. The authors said that data on Peters’ duiker is the most limited (L. 256) (some studies may be overlooked. e.g. Nakashima et al. 2018). Given that the parameters of population dynamics are also not likely to largely vary, it may be better to use data on Ogilby’s duiker also, than depending on the very limited data on Peters’ duiker. Recent studies using molecular techniques have shown that they are genetically discriminated (Johnston et al. 2011; Hassanin et al. 2012; Johnston and Anthony 2012) and may be a single species. More information should be available for Ogilby’s duiker.

Minor comments

L62. Information on duikers should be mentioned after the line 77.

L89. Recent studies using molecular techniques have revealed that Peters’s duiker

L90. Recent works use Philantomba monticola as a scientific name of blue duikers instead of Cephalophus monticola.

L121. r = exp(rmax) is correct? In Eq. 4, the authors seem to assume rmax > 0. If so, exp should be removed.

L132. This assumption influences the results. Justify this assumption here.

L142. The ‘n = 1000’ possibly means the number of iterations in the simulation. There is no need to mention here (or add the explanations).

L159. ‘n = 1000’. See above.

L256. See the major comment.

L286 Fig 4 legend. I did not find (B)

L458. I do not always agree with this statement. In central Africa, local people often capture duikers using snares. As long as the capture effort is constant, the harvest amount should be proportional to the duiker density. The results of this study seem to suggest that setting the maximum snare effort may contribute to the sustainable harvesting of duikers.

Figure 2-4 and Figure 5-7 should be integrated into one figure (with three panels), respectively.

The S9 figure largely helped readers to understand what the author actually did. I suggest that this figure moves to the main text (not supplementary).

References

Hassanin A, Delsuc F, Ropiquet A, Hammer C, van Vuuren BJ, Matthee C, Ruiz-Garcia M, Catzeflis F, Areskoug V, Nguyen TT, Couloux A (2012) Pattern and timing of diversification of Cetartiodactyla (Mammalia, Laurasiatheria), as revealed by a comprehensive analysis of mitochondrial genomes. C R Biol 335:32–50.

Johnston AR, Anthony NM (2012) A multi-locus species phylogeny of African forest duikers in the subfamily Cephalophinae: evidence for a recent radiation in the Pleistocene. BMC Evol Biol 12:120–135.

Johnston AR, Anthony NM (2012) A multi-locus species phylogeny of African forest duikers in the subfamily Cephalophinae: evidence for a recent radiation in the Pleistocene. BMC Evol Biol 12:120–135.

Nakashima Y, Fukasawa K, Samejima H (2018) Estimating animal density without individual recognition using information derivable exclusively from camera traps. J of Appl Ecol 55:735-744.

6. PLOS authors have the option to publish the peer review history of their article (what does this mean?). If published, this will include your full peer review and any attached files.

Reviewer #1: **Yes: **Robert Nasi

Reviewer #2: No

---

## [Author Response · Author response to Decision Letter 0]

3 Sep 2020

Reviewer 1

Reviewer’s summary: ‘Congratulations for a nicely designed robust study on the impact of uncertainties in the estimation of sustainable offtake for duiker species. I believe this is a very good contribution to the body of research on sustainable hunting as well as something that could be used by practitioners for other species or places.

I have no major comments to make on the paper, the methods are well explain and statistically sound to me and the conclusions reflect faithfully the findings. Useful comparisons are provided with existing studies and show the value of the proposed approach.

This is beyond the scope of the present paper but I think the next step should be to look at the "scale" issue and how hunting is done. This might have a strong influence on expected sustainable yield. You hint on these issues L178-L181 about the many ways harvesting effort can be altered and L434-441 about the issue of scale. I am wondering to which extant it would be interesting to try selecting a very well know species (e.g. roe deer) and test these....’

Response: Thank you for your positive feedback and suggestions. We have addressed them below. In this study, we have not included the question of scale focusing on the approach and the effects of incorporating uncertainty in harvesting decisions. However, we agree that the affects of scale would be very interesting to explore. With regard to harvesting effort, I did look at the effects of varying effort (both quotas and proportions) with changes in duiker population densities (a simulated passive adaptive management system) in Chapter 3 of my PhD thesis (available on https://discovery.ucl.ac.uk/id/eprint/10076541/). The results showed that iteratively adjusting harvesting quota in response to changes in density was highly beneficial in terms of meat yields and species survival. Some benefits were also present under proportional harvesting; however, proportional harvesting was to a degree ‘self-regulating’ automatically compensating for changes in density. This research has not yet been published but I plan to make it available on bioRxiv before the end of the year.

Minor comments

Reviewer: [1] is probably not the best one to quote for showing harvesting greater than six times the sustainable harvest rate. Maybe it would be better to refer to the latest analyses provided in https://doi.org/10.17528/cifor/007046

Response: Thank you, we replaced our reference [1] with the reference you suggested [2]. Your reference is more up-to-date and draws attention to the effects of sample size, and regional and site differences on harvest rates. 

Reviewer: [27,31] are a bit outdated in terms of offtakes too and more recent references could be found in https://doi.org/10.17528/cifor/007046

Response: Although some of references in https://doi.org/10.17528/cifor/007046 [2], in particular [3] could have been used to develop a different way to estimate actual offtakes per species, we feel that the two studies we used [4, 5] provided good country-level summaries and were fit for purpose. Most studies, e.g. [1], admit that estimating rates of exploitation in the Congo Basin is difficult and prone to uncertainty. Undoubtfully, this will improve with time. Please see below our comments for each of the references in [2]: 

Reference Comment

Fa et al. 2002 [6]

The results of this study were used in Fa, Currie and Meeuwig (2003) - the study we used in our manuscript.

Fa et al. 2016a [7]

Their estimate (225.7 animals km-2 yr-1) is for non-Pygmy hunters only and is based on a limited number of settlements in the region; for example, a single Pygmy village and no non-Pygmy villages in Gabon.

Ziegler et al. 2016 [3]

Although an excellent modelling study, the paper does not contain the country-specific estimates that we used to estimate species-specific offtake levels.

Ingram 2018 [8]

Although it gives an excellent comparison of different methods of estimating annual harvests across Central Africa, once again the study does not contain the country-specific estimates that we needed to estimate species-specific offtake levels.

Reviewer 2

Reviewer’s summary: ‘This paper addresses an important issue for the sustainable management of data-deficient animals. In the actual world, parameters of animal population dynamics can stochastically vary, and the uncertainty often leads to a different scenario of population dynamics. The authors explicitly incorporate the uncertainty of the parameters (r and K) in the Beverton-Holt population model and apply the model to three species of duikers in Central Africa. The authors revealed that, under quota-based harvesting, there may be a trade-off between yield and extinction probability, which was not evident when not accounting for the uncertainty of the parameters.

The manuscript is well-written and contains sufficient interest and originality to merit publication in PLOS One. However, there are several constraints and issues that should be mentioned explicitly as below.’

Response: Thank you for your positive comments and suggestions. We have addressed them below.

Major comments (please note that we used the page numbers as per Revised Manuscript in our Responses)

Reviewer: 1. The approach presented in the manuscript can be applied to any species, as mentioned in the text. I think this point should be stressed more explicitly. For example, the model details should be explained before mentioning the procedures of actual application to the duikers (in Method section). It is better to regard the application to the duikers as a case study using the more-widely applicable approaches.

Response: We agree that it is important to draw attention to the fact that our method can easily be applied to any hunted species. This is mentioned in the Introduction (L82 ) and in the Conclusion (L574). 

As you suggested, we have moved the procedures of actual application to duikers to later in the Methods section, after the model details. We have also added the following:

L88 ‘We begin by describing our modelling approach. ‘

L230 ‘as our case study’

L538 “High parameter uncertainty is common in the tropics. Although we used the duikers to illustrate our approach, the approach can be used to help inform management decisions for any harvested species.

Reviewer: 2. The approach in the manuscript still has limitations to predict bushmeat yield and extinction probability, although I admit that the authors tried to do the best within the constraints. The authors should specifically mention the limitations in the Discussion. In particular, the following two points should be mentioned.

The results of the simulations in the manuscript should largely dependent on the distribution that the parameter, r and K follow. The author assumes that the mean and sd of these parameters follow a log-normal distribution. This distribution is relatively fat-tailed, inevitably leading to a larger variance of the estimates. To make more precise predictions, it is critical to accumulate more reliable information on r and K in more fields

Response: On L528 we added “Thirdly, the values for population growth rates and carrying capacity were sampled from a log-normal distribution which has a relatively large right tail, leading to a larger variance of the estimates. More reliable estimates of population growth rates and carrying capacity are therefore critical to enable more precise predictions.”

Reviewer: � The model assumes that the population is spatially closed. However, in real world in central Africa, local people often harvest bushmeat in specific area near village within a large continuous forest. In such environments, the vacant area (i.e. the area where duikers are exhausted) should be rapidly occupied by the surrounding unhunted populations (i.e. source-sink structure). Assuming that immigration/emigration can occur, the extinction risk should be much lower than the results of the simulations”

Response: On L532 we added “Finally, we did not account for the likely replenishment of the vacant areas (i.e. areas were duikers had been exhausted) by immigrants from the surrounding unhunted populations (i.e. source-sink structure). Assuming that immigration/emigration can occur, the extinction risk should be lower than we predicted; however, given the pervading uncertainty, we recommend erring on the side of caution.”

Reviewer: The authors said that data on Peters’ duiker is the most limited (L. 256) (some studies may be overlooked. e.g. Nakashima et al. 2018). Given that the parameters of population dynamics are also not likely to largely vary, it may be better to use data on Ogilby’s duiker also, than depending on the very limited data on Peters’ duiker. Recent studies using molecular techniques have shown that they are genetically discriminated (Johnston et al. 2011; Hassanin et al. 2012; Johnston and Anthony 2012) and may be a single species. More information should be available for Ogilby’s duiker.

Response: Thank you for bringing to our attention that C.callipygus and C.ogilbyi may be subspecies. A number of studies in our dataset, e.g. [9, 10] separate the two species providing disparate species estimates. Therefore, we feel uncomfortable combining the estimates of these two possible subspecies in our dataset. Including additional estimates may also increase variability rather than decrease it. Our approach can account for larger parameter variability, and by providing a detailed description of the method, we encourage readers to do so.

Minor Comments

Reviewer: L62. Information on duikers should be mentioned after the line 77.

Response: We moved information about the duikers to the final paragraph (after L77) as recommended.

Reviewer: L89. Recent studies using molecular techniques have revealed that Peters’s duiker 

Response: Unfortunately, the end of the reviewer’s comment got lost. But we think it was about Peters’s duiker C.callipygus and Ogilby’s duiker C.ogilbyi possibly being subspecies. We address this point in the Major comments above.

Reviewer: L90. Recent works use Philantomba monticola as a scientific name of blue duikers instead of Cephalophus monticola.

Response: We have kept the name Cephalophus monticola in the manuscript as this name was used by the authors included in our analysis. We have also added the following comment: L195 'Three Cephalophus species: Peters’ duiker C. callipygus, bay duiker C.dorsalis and blue duiker C. monticola (also known under the scientific name Philantomba monticola) [11, 12]...’

Reviewer: L121. r = exp(rmax) is correct? In Eq. 4, the authors seem to assume rmax > 0. If so, exp should be removed.

Response: Removed exp, thank you.

Reviewer: This assumption influences the results. Justify this assumption here 

Response: L105 We have added the following “with 0.10 being the lowest value implemented by Lande, Sæther and Engen [13] reflecting low climate variability in the tropics.”

Reviewer: L142. The ‘n = 1000’ possibly means the number of iterations in the simulation. There is no need to mention here (or add the explanations).

Response: Removed ‘n = 1000’ as suggested; the number of iterations (1000) is given on line 174.

Reviewer: ‘n = 1000’. See above. 

Response: Removed ‘n = 1000’ as suggested; the number of iterations (1000) is given on line 174.

Reviewer: ‘L256. See the major comment.

Response: Covered in the Major Comments.

Reviewer: L286 Fig 4 legend. I did not find (B).

Response: Removed reference to (B), thank you. (B) was left over from when the figure combined cases with and without uncertainty.

Reviewer: L458. I do not always agree with this statement. In central Africa, local people often capture duikers using snares. As long as the capture effort is constant, the harvest amount should be proportional to the duiker density. The results of this study seem to suggest that setting the maximum snare effort may contribute to the sustainable harvesting of duikers. 

Response: L470 We added “In principle, harvesting could be limited to a proportion of animal densities by, for example, keeping the number of snares constant. However, this is not always feasible due to poor harvesting regulation in West and Central Africa.” 

On L464 we do admit that proportional harvesting is not perfect and the policy that maximises yield still results in some extinctions. On L466 we have added “Therefore, we caution against using the maximum target proportion.”

Reviewer: Figure 2-4 and Figure 5-7 should be integrated into one figure (with three panels), respectively.

Response: In terms of combining multiple figures, we are constrained by the PLOS One’s maximum figure size requirements and image resolution. We have tried combining multiple plots; however, the resulting figures were not as legible, in particular, survival probabilities were too small to read.

Reviewer: The S9 figure largely helped readers to understand what the author actually did. I suggest that this figure moves to the main text (not supplementary). 

Response: We have moved Figure S9 to the ‘Framework Summary’ section. It is now Fig 1.

 

References:

1. Fa JE, Olivero J, Farfán MA, Lewis J, Yasuoka H, Noss A, et al. Differences between Pygmy and Non-Pygmy Hunting in Congo Basin Forests. PLoS One. 2016;11(9):e0161703-e.

2. Coad L, Fa JE, Abernethy K, Van Vliet N, Santamaria C, Wilkie D, et al. Towards a sustainable, participatory and inclusive wild meat sector: CIFOR; 2019.

3. Ziegler S, Fa JE, Wohlfart C, Streit B, Jacob S, Wegmann M. Mapping bushmeat hunting pressure in Central Africa. Biotropica. 2016;48(3):405-12.

4. Fa JE, Currie D, Meeuwig J. Bushmeat and food security in the Congo Basin: linkages between wildlife and people's future. Environmental Conservation. 2003;30(1):71-8.

5. Wilkie DS, Carpenter JF. Bushmeat hunting in the Congo Basin: an assessment of impacts and options for mitigation. Biodiversity {&} Conservation. 1999;8(7):927-55.

6. Fa JE, Peres CA, Meeuwig J. Bushmeat exploitation in tropical forests: an intercontinental comparison. Conservation Biology. 2002;16(1):232-7.

7. Fa JE, Olivero J, Farfán MA, Lewis J, Yasuoka H, Noss A, et al. Differences between Pygmy and Non-Pygmy Hunting in Congo Basin Forests. PloS one. 2016;11(9):e0161703-e.

8. Ingram DJ. Quantifying the exploitation of terrestrial wildlife in Africa. 2018.

9. Fa JE, Seymour S, Dupain JEF, Amin R, Albrechtsen L, Macdonald D. Getting to grips with the magnitude of exploitation: bushmeat in the Cross–Sanaga rivers region, Nigeria and Cameroon. Biological Conservation. 2006;129(4):497-510.

10. Coad LM. Bushmeat hunting in Gabon: socio-economics and hunter behaviour. 2008.

11. Johnston AR, Anthony NM. A multi-locus species phylogeny of African forest duikers in the subfamily Cephalophinae: evidence for a recent radiation in the Pleistocene. BMC Evol Biol. 2012;12:120-.

12. van Vliet N, Nasi R. What do we know about the life-history traits of widely hunted tropical mammals? Oryx. 2019;53(4):670-6.

13. Lande R, Sæther B-E, Engen S. Threshold harvesting for sustainability of fluctuating resources. Ecology. 1997;78(5):1341-50.

---

## [Editor Report · Decision Letter 1]

14 Sep 2020

Modelling parameter uncertainty reveals bushmeat yield versus survival trade-offs in heavily-hunted duiker Cephalophus spp.

PONE-D-20-16287R1

Dear Dr. Barychka,

We’re pleased to inform you that your manuscript has been judged scientifically suitable for publication and will be formally accepted for publication once it meets all outstanding technical requirements.

Kind regards,

Stephanie S. Romanach, Ph.D.

Academic Editor

PLOS ONE
---

## [Editor Report · Acceptance letter]

18 Sep 2020

PONE-D-20-16287R1 

Modelling parameter uncertainty reveals bushmeat yields versus survival trade-offs in heavily-hunted duiker *Cephalophus* spp. 

Dear Dr. Barychka:

I'm pleased to inform you that your manuscript has been deemed suitable for publication in PLOS ONE. Congratulations! Your manuscript is now with our production department. 

Kind regards, 

on behalf of

Dr. Stephanie S. Romanach 

Academic Editor

PLOS ONE